# Mechanical Properties of Cemented Particulate Composite: A 3D Micromechanical Model

**DOI:** 10.3390/ma14143875

**Published:** 2021-07-12

**Authors:** Chenglin Tao, Xi Liang, Xiaoxue Bi, Zeliang Liu, Huijian Li

**Affiliations:** 1Kcy Laboratory of Mechanical Reliability for Heavy Equipment and Lark Structures of Hebei Province, Yanshan University, Qinhuangdao 066004, China; taochenglin@stumail.ysu.edu.cn (C.T.); liuzeliang@ysu.edu.cn (Z.L.); 2School of Civil Engineering and Mechanics, Yanshan University, Qinhuangdao 066004, China; lxzhouxin@126.com

**Keywords:** cemented particulate composite, damage evolution process, 3D rigid beam-spring, stiffness matrix

## Abstract

Cemented particulate composite is a kind of composite material with high strength, high energy absorption, and multifunctional characteristics, which is widely used in engineering practice. The calculation of the mechanical properties of granular composites based on theoretical methods has always been a topic of discussion. A micromechanical model with a three-dimensional rigid beam-spring network (3D-RBSN) is proposed here. The stiffness matrix of the model was calculated theoretically. The model was applied to the analysis of the mechanical properties of composites material with glass beads and epoxy resin. The results indicate that the 3D-RBSN model can effectively predict the mechanical properties of composite materials, such as Young’s modulus and Poisson’s ratio. Furthermore, the damage evolution process of cemented particulate composite with initial defects was analyzed based on the 3D-RBSN model.

## 1. Introduction

Particle-reinforced composite material is made of metal or non-metal material and is prepared by compounding ceramic, glass, metal, and other particles as a reinforcing phase [1,2]. The particle-reinforced composite material combines the properties of the matrix and the reinforcement and has the advantages of high strength, high rigidity, stable mechanical properties, and easy processing into a specific structural form. Therefore, it is widely used in many fields such as mechanical construction, chemical aerospace, and other similar areas [3,4,5]; for example, aggregate concrete, iron-based alloy, open-cell metal hollow spherical foam, and other particle reinforced composite materials [6].

In the past few decades, many micromechanical model research methods have been developed to analyze the effective mechanical properties of particle-reinforced composites employing effective medium theory. There is the traditional inclusion theory, the Mori–Tanaka method [7], the self-consistent method [8], the generalized self-consistent method [9], the representative volume element method [10], the differential method [11], the multiphase spherical model method [12] and so on. However, due to a large number of voids in the cemented particle composite, there are typically discontinuous mechanical properties and these methods have certain limitations.

To solve the above problems, Zhou Chunwei [13,14,15] proposed a meso-mechanical model of high volume content particle-reinforced composites, which simplified the particles into homogeneous elastic spheres of the same size, and the connecting matrix between the two particles into a short elastic-plastic cylinder. Assuming that the mesoscopic stress, strain, and plastic zone are axisymmetric distributions, the integral equations reflecting the normal and tangential deformation of a pair of particles are established, and the macroscopic stress-strain relationship of the material is deduced. Hrennikoff et al. [16,17,18,19] equated particle composites as network models and successfully solved many elastic mechanics problems. Subsequently, this model was used to simulate the fracture process of particle composites. Liu Jinxing [20] improved the beam mesh fracture model of particle composites and obtained a reasonable force-displacement relationship and fracture mode by using the improved model. Eckschlager [21,22] proposed a continuous failure model of brittle particles based on the finite element method. The damage caused by the fracture of particles was simulated by the uniaxial tensile load. The results of continuous solutions covering multiple particles were given and discussed from two aspects of macro-response and evolution of fracture probability of particles. However, the above studies did not consider the bending conditions of the model and could not accurately calculate the macroscopic elastic-plastic properties or accurately describe the damage evolution process.

Glass bead/epoxy resin-cemented particle composite is a new type of structural particle-reinforced composite, which has been widely used in aerospace, national defense, construction, transportation, and other industries [23]. In terms of composition and structure, the epoxy resin has a close molecular structure, strong cohesion, and strong adhesion, which can be combined well with glass beads and form a stable composite. At the same time, the addition of glass beads endows glass beads/epoxy resin with high strength, energy absorption, and shock absorption. At present, the research on glass bead/epoxy composites is more focused on the preparation process and conventional mechanical properties, such as compression, water absorption, specific strength, and so on. For example, Li Chune [24] analyzed the curing reaction of an epoxy resin/glass bead system with different temperatures and different contents of glass bead by using non-equilibrium thermodynamic fluctuation theory. Zhou Jinlei [25] analyzed the activation energy, water absorption, and compressive strength of hollow glass beads and epoxy resin composites. Yang Qingquan [26] analyzed the thermal energy in the glass bead/epoxy resin curing system and the mechanical properties and water absorption after curing. Wang Jian et al. [27] discussed the effects of glass beads added into epoxy resin on its stiffness, strength stability, insulation, and other properties. Yang Zhiqiang [28] applied the properties of epoxy resin and glass beads, added carbon nanotubes and carbon fibers, and made composites with higher strength and greater modulus. Yang Qinghai et al. [29] studied the density, pressure, and failure mode of glass bead/epoxy resin composites. Hu Chuanqun et al. [30] studied the tensile and impact properties of modified glass bead/epoxy resin composites.

Based on the above meso-mechanical model and beam network model, Liang [31] proposed a new theory of composite spherical elements (CSE). The theory combines the mesoscopic mechanics analysis method with the macro-analysis method of the grillage model, considers the bending stiffness of the CSE, equates the CSE with the rigid beam-spring model and assembles it into a plane network model, effectively calculates the elastic properties of the granular composite material, and simulates the fracture damage of the material by using the finite element software.

The above study only equates the material into a two-dimensional planar structure. However, according to existing research [32,33], the two-dimensional (2D) model cannot reflect the true structure of the material and often gives an unsatisfactory prediction of the mechanical response of the particle reinforcement. In this paper, based on the theory of CSE, the torsional stiffness of the structure was calculated and the stiffness matrix of 3D CSE is given. The CSE was equivalent to a 3D rigid beam-spring model, and an equivalent 3D network model describing the arrangement of particles in the material was further established. The elastic modulus and Poisson’s ratio of glass bead/epoxy resin-bonded composites were obtained by numerical simulation using a finite element method. The damage evolution process of bonded cemented particulate composite with initial defects was given.

## 2. 3D Micromechanical Model

### 2.1. Calculation of Torsional Stiffness of CSE

Cemented particle composite is a combination of particles and particles connected by a matrix. The particles are not tightly connected and there are a large number of voids, which is a composite material containing pores. Figure 1 shows two kinds of cemented particulate composite. One is a hollow glass microbead composite (as shown in Figure 1a) and the other is sandstone (SEM micrograph) (as shown in Figure 1c).

Liang [31] proposed a CSE model based on the special structure of cemented particulate composite. The model considers a pair of adjacent particles in the material and the matrix between them as a whole, and this whole structure is called a CSE, as shown in Figure 1b. The CSE model can be used to analyze cemented composites from a microscopic to a macroscopic perspective. By connecting the centers of two adjacent particles in a cemented composite with a straight line, the entire material can be equivalent to a beam network element model, as shown in Figure 2a,b. According to the method of solving the beam network model, a CSE is regarded as a beam element in the beam network model. Based on the elastic mechanics theory and the finite element method, the stiffness expression of the CSE under tension, shear, and bending was obtained. However, these three kinds of stiffness can only describe the two-dimensional model. To accurately describe the mechanical properties of the combined ball unit, it was necessary to push down the torsional stiffness of the combined ball unit based on the two-dimensional model.

Based on CSE theory, the torsional stiffness of CSE was obtained by calculation. Figure 3 is a schematic view showing deformation of a twisting force moment applied to both ends of a CSE. It was assumed that the deformation region of the particle was an elastic half-space. For the particle, the position where the material was deformed when the material was stressed was only the contact area of the particle with the substrate and its vicinity, so the equatorial plane of the two particles was considered to remain flat. In addition, the axial symmetry plane of the CSE did not produce bending deformation, and the equatorial planes of the two particles only rotated relative to each other without axial displacement.

For a single CSE, a coordinate system was established as shown in Figure 4. Among them, two spheres were symmetrical to the x-z plane. The y-axis coincided with the line connecting the two spheres. R and $r_{c}$ are sphere radius and matrix radius, respectively. The vertical length between the point at any position in the matrix and the y axis is r. The minimum thickness of the substrate at the y-axis is $h_{0}$, and the maximum thickness is $h_{c}$. The thickness at any position can be expressed by a function of h(r).

When the CSE was subjected to a torsion, the force equation was constructed. Assuming that a torque T was applied to a CSE alone, the relative angular angle of the two equatorial planes was α. Then, the matrix was not subject to axial stress, only the tangential force, the tangential force was only related to the radius r, and its value was symmetric with respect to r=0. Let the stress function be a quartic polynomial of *r*. Considering the above characteristics of shear stress distribution, A is the undetermined stress coefficient, the function form is set as:(1)$$\tau \left(r\right)=Ar^{4}$$

According to reference [31], the axial and tangential deformation of matrix and particle can be expressed as follows:(2)$$\left\{\begin{array}{l} u_{m}\left(r,\theta \right)=\frac{\left(1-2u_{m}\right)}{4\left(1-u_{m}\right)G_{m}}h\left(r,\theta \right)\delta_{n}\left(r,\theta \right)\\ u_{p}\left(r,\theta \right)=\frac{\left(1-u_{p}\right)}{2\pi G_{p}}\int_{0}^{\pi }d\phi \int_{0}^{r\cos\phi +\sqrt{r_{c}^{2}-r^{2}\sin^{2}\phi }}\times \sigma_{n}\left(\sqrt{r^{2}+s^{2}-2rs\cos\phi }\right)ds \end{array}\right.$$
(3)$$\left\{\begin{array}{l} v_{m}\left(r,\theta \right)=\frac{1}{2G_{m}}h\left(r,\theta \right)\tau_{nx}\left(r,\theta \right)\\ v_{p}\left(r,\theta \right)=\frac{1}{2\pi G_{p}}\int_{0}^{\pi }d\phi \int_{0}^{r\cos\phi +\sqrt{r_{c}^{2}-r^{2}\sin^{2}\phi }}\times \tau_{nx}\left(\sqrt{r^{2}+s^{2}-2rs\cos\phi }\right)\left(1-u_{p}\sin^{2}\phi \right)ds \end{array}\right.$$
where $G_{m}$, $G_{p}$, $\mu_{m}$ and $\mu_{p}$ are shear modulus and Poisson’s ratio of matrix and particle, respectively.

The distribution coefficient A in the formula (1) was calculated by the point collocation method: we selected a point on the boundary between the matrix and the particle cementation surface. At point $r=r_{c}$:(4)$$\begin{array}{l} v_{m}=\frac{h_{c}Ar^{4}}{2G_{m}}\\ v_{p}=\frac{A}{2\pi G_{P}}\times \left(\frac{0.79}{r_{c}{}^{5}}-0.392\mu_{p}\right) \end{array}$$
where, $v_{m}$ is the tangential displacement of any point on the contact interface relative to the x-z plane, namely the displacement of the matrix surface. $v_{p}$ is the tangential motion of the point and the equatorial plane of the particle, namely the displacement of the particle. $h_{c}$ is the thickness at the edge of the matrix.
(5)$$\frac{\alpha }{2}r_{c}=v_{m}\left|{}_{r=r_{c}}\right.+v_{p}\left|{}_{r=r_{c}}\right.$$

Formulas (4) and (5) were substituted into Formula (6):(6)$$\frac{\alpha }{2}r_{c}=c_{1}A+c_{2}A$$

In the formula, $c_{1}$ and $c_{2}$ are undetermined coefficients.

The relationship between torque and deformation of particles in elastic stage is as follows.
(7)$$T=\int_{0}^{2\pi }\int_{0}^{r_{c}}\tau \left(r\right)r^{2}drd\theta =K_{T}\alpha$$

In the formula, $K_{T}$ is the torsional rigidity of the CSE. Equation (7) results in
(8)$$T=c_{3}A$$
where $c_{3}$ is the undetermined coefficient.

According to Equations (6)–(8):(9)$$c_{1}=\frac{h_{c}r_{c}{}^{4}}{2G_{m}}$$
(10)$$c_{2}=\frac{1}{2\pi G_{p}}\left(0.791r_{c}{}^{5}-0.392\mu_{p}\right)$$
(11)$$c_{3}=\frac{2\pi }{7}r_{c}{}^{7}$$

Then, the shear stress function is:(12)$$\tau \left(r\right)=\frac{r^{4}}{c_{3}}T$$
obtained from $T=K_{T}\alpha$:(13)$$K_{T}=\frac{c_{3}r_{c}}{2\left(c_{1}+c_{2}\right)}$$

In the equation, the expressions of $c_{1}$, $c_{2}$ and $c_{3}$ are shown in Equations (9)–(11), which are related to the mechanical parameters of materials and the structural dimensions of single CSE.

### 2.2. Establishment of CSE Stiffness Matrix

The element stiffness matrix of the CSE was constructed according to the finite element method. A pair of spheres connected by a composite matrix was considered as a unit in which two spherical centers i and j were regarded as element nodes. As shown in Figure 5, a local coordinate system was established at the two spherical centers of the CSE. The origin of the coordinate system was at the spherical center and the directions of the two coordinate systems were identical. The centers of the two ends of the space CSE were nodes i and j, respectively. Each node had six node displacement components and six node force components. The node variables of the six degrees of freedom CSE are given in the figure. $u_{i}$, $v_{i}$, $w_{i}$, $\theta_{xi}$, $\theta_{yi}$, $\theta_{zi}$ and $u_{j}$, $v_{j}$, $w_{j}$, $\theta_{xj}$, $\theta_{yj}$, $\theta_{zj}$ denote the displacement and torsion angle of ball i and j, respectively. The arrows point in the positive direction. $X_{i}$, $Y_{i}$, $Z_{i}$, $M_{xi}$, $M_{yi}$, $M_{zi}$ and $X_{j}$, $Y_{j}$, $Z_{j}$, $M_{xj}$, $M_{yj}$, $M_{zj}$ represent the force and moment of force caused by the displacement of the spherical centers i and j of the CSE.

The direction of moment and angular displacement of the couple is expressed by double arrows according to the right-hand rule, and the direction of force and line displacement is expressed by single arrows. The arrows shown in Figure 5 are all positive.

Figure 6a–f shows the nodal force of the spatial CSE caused by the displacement of the unit node. All directions indicated in the figure are in the positive direction. First, we obtained the force of the center of the sphere when the component of $\delta^{e}$ was 1, while the rest was 0. The relation between the force acting on the center of the sphere and the displacement of the center of the sphere was shown when the unit displacement occurs at the i of the element e. The force acting on the center of the sphere and the displacement component of the center of the sphere are not shown in the figure, and in this case, the force acting on the center of the sphere was zero. This method can be used to determine the relationship between the force acting on the spherical center and the displacement of the spherical center when the unit displacement of the element occurs.

Based on the superposition principle, the matrix expressions of nodal displacement and nodal force of CSE can be obtained as follows:(14)$$F^{e}=K^{e}\delta^{e}$$
where $F^{e}$ represents the nodal force, $K^{e}$ is the element stiffness matrix, and $\delta^{e}$ is the displacement matrix. The direct expression of the finite element method was used to construct the expressions of the CSE as follows:(15)$$F^{e}={\left[\begin{array}{cccccccccccc} X_{i} & Y_{i} & Z_{i} & M_{xi} & M_{yi} & M_{zi} & X_{j} & Y_{j} & Z_{j} & M_{xj} & M_{yj} & M_{zj} \end{array}\right]}^{T}$$
(16)$$K^{e}=[\begin{array}{cccccccccccc} K_{U} & 0 & 0 & 0 & 0 & 0 & -K_{U} & 0 & 0 & 0 & 0 & 0\\ 0 & K_{V} & 0 & 0 & 0 & \frac{1}{2}lK_{V} & 0 & -K_{V} & 0 & 0 & 0 & -\frac{1}{2}lK_{V}\\ 0 & 0 & K_{V} & 0 & -\frac{1}{2}lK_{V} & 0 & 0 & 0 & -K_{V} & 0 & -\frac{1}{2}lK_{V} & 0\\ 0 & 0 & 0 & K_{T} & 0 & 0 & 0 & 0 & 0 & -K_{T} & 0 & 0\\ 0 & 0 & -\frac{1}{2}lK_{V} & 0 & \frac{1}{4}l^{2}K_{V}+K_{M} & 0 & 0 & 0 & \frac{1}{2}lK_{V} & 0 & \frac{1}{4}l^{2}K_{V}-K_{M} & 0\\ 0 & \frac{1}{2}lK_{V} & 0 & 0 & 0 & \frac{1}{4}l^{2}K_{V}+K_{M} & 0 & -\frac{1}{2}K_{V} & 0 & 0 & 0 & \frac{1}{4}l^{2}K_{V}-K_{M}\\ -K_{U} & 0 & 0 & 0 & 0 & 0 & K_{U} & 0 & 0 & 0 & 0 & 0\\ 0 & -K_{V} & 0 & 0 & 0 & -\frac{1}{2}lK_{V} & 0 & K_{V} & 0 & 0 & 0 & -\frac{1}{2}lK_{V}\\ 0 & 0 & -K_{V} & 0 & \frac{1}{2}lK_{V} & 0 & 0 & 0 & K_{V} & 0 & \frac{1}{2}lK_{V} & 0\\ 0 & 0 & 0 & -K_{T} & 0 & 0 & 0 & 0 & 0 & K_{T} & 0 & 0\\ 0 & 0 & -\frac{1}{2}lK_{V} & 0 & \frac{1}{4}l^{2}K_{V}-K_{M} & 0 & 0 & 0 & \frac{1}{2}K_{V} & 0 & \frac{1}{4}l^{2}K_{V}+K_{M} & 0\\ 0 & \frac{1}{2}lK_{V} & 0 & 0 & 0 & \frac{1}{4}l^{2}K_{V}-K_{M} & 0 & -\frac{1}{2}lK_{V} & 0 & 0 & 0 & \frac{1}{4}l^{2}K_{V}+K_{M} \end{array}]$$
(17)$$\delta^{e}={\left[\begin{array}{cccccccccccc} \mu_{i} & \nu_{i} & \omega_{i} & \theta_{xi} & \theta_{yi} & \theta_{zi} & \mu_{j} & \nu_{j} & \omega_{j} & \theta_{xj} & \theta_{yj} & \theta_{zj} \end{array}\right]}^{T}$$

Therefore, we used the 3D model after adding torsion stiffness to correct the Equation (18) of normal stress and shear stress obtained from the two-dimensional model in reference [31]. Among them, A, $A_{1}$, $A_{2}$, $B_{1}$, $B_{2}$, $B_{3}$ are the stress distribution coefficients. If the node displacement of a CSE was known, the stress of the element could be obtained. The expressions of normal stress and shear stress were revised, respectively:(18)$$\left\{\begin{array}{l} \sigma \left(r,\theta \right)=A_{1}r^{2}+B_{1}+A_{3}r^{3}\cos^{3}\theta +B_{3}r^{3}\cos\theta \sin^{2}\theta \\ \tau \left(r,\theta \right)=A_{2}r^{2}+B_{2}+Ar^{4} \end{array}\right.$$

### 2.3. 3D Rigid Beam-Spring Equivalent Model

According to the numerical simulation method of composite micro-mechanics [34,35], the spatial structure of a CSE was compared with that of a beam element, and the element stiffness matrix of the two elements was similar. Therefore, the degree of freedom of each node of the spatial CSE was like the beam element, which is the degree of freedom of movement in the three directions of the x-axis, the y-axis, and the z-axis, and the degree of freedom of rotation about each axis. Depending on the stiffness of the ball in all directions of the unit, a single CSE could be described by a rigid beam-spring simplified model as shown in Figure 7. Each node in the equivalent rigid beam-spring model also had six degrees of freedom.

It was assumed that the deformation of CSE under load was mainly concentrated in the bonding part of particles and matrix and its vicinity, while no deformation occurred in other areas. Therefore, the part without deformation was regarded as a rigid body, and only the material between the lateral symmetry planes of the two particles was taken in the calculation.

In Figure 7, the simplified model consists of four longitudinal springs, two transverse springs, and one axial torsional spring. To make the simplified model and the CSE have the same effect of stiffness in all directions, the elastic coefficients of the four longitudinal springs were set to $K_{U}/4$, The elastic coefficients of the two transverse springs were set to $K_{v}$. Adding torsion springs between spherical centers i and j at both ends of the equivalent model, the elastic coefficient of the torsion spring was $K_{M}$. The torsion effect was the same as that of the CSE model. l represents the distance between the transverse symmetry planes of the two particles in the CSE, that is, the distance between the simplified equivalent model node i and j. The relative distance between the two tension springs is d. The setting of d ensured that the simplified model and the CSE had the same bending stiffness. The value of d was calculated as follows:(19)$$d=\sqrt{\frac{2K_{M}}{K_{U}}}$$

The particles inside the cemented composite material were arranged in a plurality of ways, two adjacent particles cemented together in the composite material were joined to form a CSE, and a pair of CSEs cemented together were arranged at different angles, wherein the distance between the centers of any pair of particles was l. In this way, the granular composite material model constructed by connecting multiple CSEs could accurately describe its internal structure. In this paper, two arbitrary adjacent CSE were arranged in 90 (regular hexahedron arrangement) and 60° (regular tetrahedron arrangement), as shown in Figure 8a,b: schematic diagrams of cement particle composites with two different arrangement forms and their internal particle arrangement diagrams, respectively.

Then, the whole composite material was similar to a beam network model. A rigid beam-spring element was regarded as a beam element in the beam network model, so that a network model composed of several rigid beam-spring elements could be equivalent to the mechanical properties of the whole composite material. Each element was connected by nodes, and the nodes of the rigid beam-spring composite element model coincided with the position of the spherical center of each particle. As shown in Figure 9a,b are spatially equivalent 3D-RBSN models with two different particle arrangements and their internal structural models.

According to the equivalent 3D rigid beam-spring network model, the finite element analysis software can be used to model the cemented particle composite material and the micromechanical analysis of the material. Compared with the planar network model in the literature [31], the reference is a two-dimensional equivalent model, which can only be used to describe the plane structure. In this paper, a three-dimensional model of combined spherical element was established, which is more consistent with the actual situation; this model can more accurately simulate the mechanical properties of 3D materials, more realistically reflect the spatial arrangement of cemented particulate composite, and better meet the mechanical properties analysis of the actual application of materials.

## 3. Epoxy Resin Glass Beads Mechanical Properties

### 3.1. Materials and Methods

Based on the 3D-RBSN model established in this paper, the mechanical properties of cemented particulate composite were studied. The particle composite material selected in this paper was glass microbead/epoxy resin, and the glass beads had the same particle size. The elastic modulus of the selected resin and glass microspheres were E = 3 GPa and E = 70 GPa, respectively; their Poisson’s ratio was u = 0.5, u = 0.35; the radius of the selected glass microspheres was R = 1.5 mm; the center-to-space spacing of the two particles in the composite material was h = 0.25 R; and the bond radius of the resin matrix in the middle of the two particles was r = 0.5 R. Substituting the material parameters into Equations (11) and (12) gave the stress distribution function of the CSE under unit torsional load:(20)$$\tau \left(r\right)=8.35r^{4}=8.35\left(x^{4}+z^{4}+2x^{2}z^{2}\right)$$

Substituting the material parameters into Equation (13) gave the value of torsional stiffness under unit torsional loading:(21)$$K_{T}=\frac{c_{3}r_{c}}{2\left(c_{1}+c_{2}\right)}=0.403\times 10^{3}N•mm/rad$$

Referring to the stiffness values of tension, compression, shear, and bending under unit load given in reference [23], the four stiffnesses of the above-mentioned CSE were taken as the parameters of the four kinds of springs in the rigid beam-spring model and the mechanical properties of the granular composites were simulated and calculated by the finite element software. In this paper, two examples of 3D-RBSN models equivalent with different particle arrangement modes of regular hexahedron and regular tetrahedron were given. Since the material was composed of many particles, if a unidirectional load was applied, the boundary effect of the particles on the boundary of the material had little effect on the entire material and was negligible, so we directly applied the equivalent node load. By applying an axial force F to the material, the macroscopic stress of the material could be obtained by dividing the axial force by the cross-sectional area, and the macroscopic elastic modulus and macroscopic Poisson’s ratio of the material could be calculated by calculation. Using the finite element software ANSYS to apply a fixed constraint on the bottom surface of the space network model, and applying the upward nodal force on the top node, the elastic modulus and Poisson’s ratio of the two different arrangements could be obtained by calculation.

As a comparison, the finite element results of the representative volume element (RVE) model were compared with those of CSE. The method of RVE is to simplify the material into a representative unit by using symmetry, and apply symmetric constraints on the partition surface, fixed constraints on the bottom surface, and uniform load on the top surface. In Figure 10a,b, the finite element models of the RVE method for the hexahedron layout and the RVE method for the tetrahedron layout are presented.

### 3.2. Results and Discussion

Through the ANSYS software simulation calculation, Table 1 shows the elastic modulus and Poisson’s ratio of the composite materials under two different arrangements calculated by the CSE and RVE methods, respectively.

By comparing the elastic modulus and Poisson’s modulus of composites with two different arrangements in Table 1, it was concluded that the mechanical properties of the composites with particles arranged in the tetrahedron were better than those of the composites arranged in the hexahedron. Comparing the CSE method and RVE method, the difference between the elastic modulus and Poisson’s ratio obtained by the two methods was within 5%. It can be seen that the CSE method can calculate the mechanical properties of granular composites more accurately, and the 3D-RBSN model can accurately reflect the mechanical properties of the CSE. Compared with the two-dimensional model in reference [31], which can only describe the plane structure, the three-dimensional model can describe the mechanical properties of materials more accurately. It can be seen from the chart data that the elastic modulus of the tetrahedral arrangement was greater than that of the hexahedral arrangement, indicating that the former has greater rigidity and better mechanical properties.

### 3.3. Damage Evolution Process of Materials with Defects

There must be slight defects in cemented particulate composite in the actual production process; the essence is the absence of particles or degumming between particles and matrix. As shown in Figure 11, a model cross-sectional view of a spatial regular hexahedron arrangement was taken as an example and the main defects were the loss of edge particles of the material and the lateral and longitudinal degumming between the particles and the particles, the absence of particles inside the material, and the lateral and longitudinal debonding between the particles. When the material was subjected to an external load, the stress concentration occurred at the defect of the material. If the local stress was too large, the material would be first destroyed at the defect, and the matrix fracture or debonding between the particles and the matrix would occur. With the continuous application of external loads, the damage would increase, and then macro-crack or fracture would occur. The traditional representative voxel method has a large amount of calculation and high requirement for computing equipment, which makes it difficult to realize in the calculation process of a large-scale supercell model. A random damage model of materials was established by randomly setting a certain number of defects on the 3D-RBSN model established in the previous paper. The damage model was simulated by the finite element software ANSYS, and the effects of particle loss and degumming on material damage and the fracture damage evolution process of the material were obtained.

For a single CSE, the CSE would be damaged when subjected to large normal stress or shear stress. Studies have shown that for the epoxy resin materials used in this paper, the tensile failure limit is about three times that of shear failure. Therefore, the judgment basis for specifying the destruction of the CSE was: (1) the CSE with the largest axial stress underwent tensile fracture in the composite 3D network model; and (2) shear fracture occurred in a composite 3D network model in which the CSE with shear stress greater than one-third of the maximum axial stress occurred. The random function was used to set up a certain number of defects in the 3D network model, that is, to remove a certain number of composite spherical rigid beam-spring models. Figure 12 shows a schematic diagram of CSE elements randomly removed from the rigid beam-spring network model. Among them, (a) shows the internal broken CSE element schematic of regular hexahedron arrangement and (b) shows the internal broken CSE element schematic of regular tetrahedron arrangement.

The ANSYS software was used to apply a uniform tensile force to the 3D network model after the defect was set, and the model was quasi-statically stretched. A new model was obtained by removing the CSE with the largest axial stress or the CSE whose shear force exceeded one third of the maximum axial force. This loading process was repeated until the entire model was completely broken. After successively applying force and removing the destroyed unit, the damage evolution process of the material could be simulated. The damage evolution paths of two models with different arrangement of elements are shown in Figure 13. Among them, (a) shows the damage evolution process of regular hexahedron arrangement materials and (b) shows the damage evolution process of regular tetrahedron arrangement materials.

Results obtained from the ANSYS software analysis showed that in the process of cracking, the failure of transverse CSE was caused by shear force, while the failure of vertical or 60 degree oblique CSE was caused by axial force. When there were a large number of random defects in the material, the cross-sectional shape formed by the different arrangement of the two particles was basically similar, the material cross-section was affected by the internal defects, and the material cross-section fluctuated with the defects, forming up-and-down wave sections along the horizontal direction, which accurately reflected the damage evolution process of the material. Compared with the section in [31], the crack propagation path of the 2D rigid beam-spring network model depended largely on the initial crack setting. The 3D rigid beam-spring network model in this paper can more accurately reflect the damage evolution path of the material.

## 4. Conclusions

In summary, a rigid beam-spring model was established based on the obtained 3D stiffness matrix of the CSE. The 3D-RBSN model and the traditional typical voxel method were used to simulate the mechanical properties of composites. The results show that the elastic modulus and Poisson’s ratio of the composite material obtained by the two methods were similar. This suggests that the established 3D-RBSN model can accurately predict the elastic properties of cemented particulate composite. The study of the mechanical properties of the bonded composites with different arrangements showed that the tetrahedral arrangement can obtain a larger Young’s modulus and Poisson’s ratio than the hexahedral arrangement.

This was compared with the two-dimensional equivalent model in reference [31], which can only be used to describe the plane structure. The three-dimensional model of combined spherical element established in this paper is more in line with the actual situation and can more accurately describe the mechanical properties of materials.

The damage evolution was analyzed by studying the structural reconstitution and mechanical properties of composites with random defects. The results indicate that the 3D-RBSN model can accurately reflect the damage evolution process of cemented particulate composite with random initial defects. It has a certain guiding effect on predicting the damage and fracture of cemented particulate composite.

Overall, the 3D-RBSN model proposed in this paper can effectively predict the mechanical properties and fracture of cemented particle composite.

## Figures and Tables

**Figure 1 materials-14-03875-f001:**
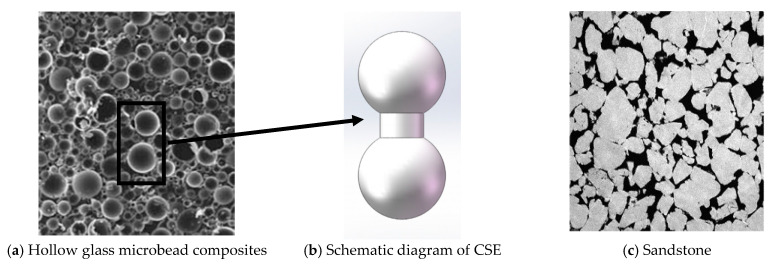
The meso-structure of cemented particulate composite.

**Figure 2 materials-14-03875-f002:**
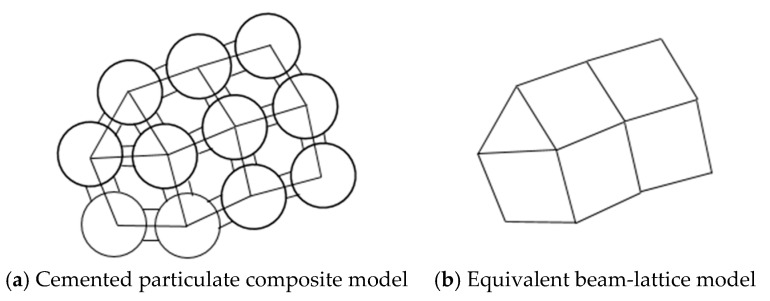
Schematic diagram of equivalent beam network model.

**Figure 3 materials-14-03875-f003:**
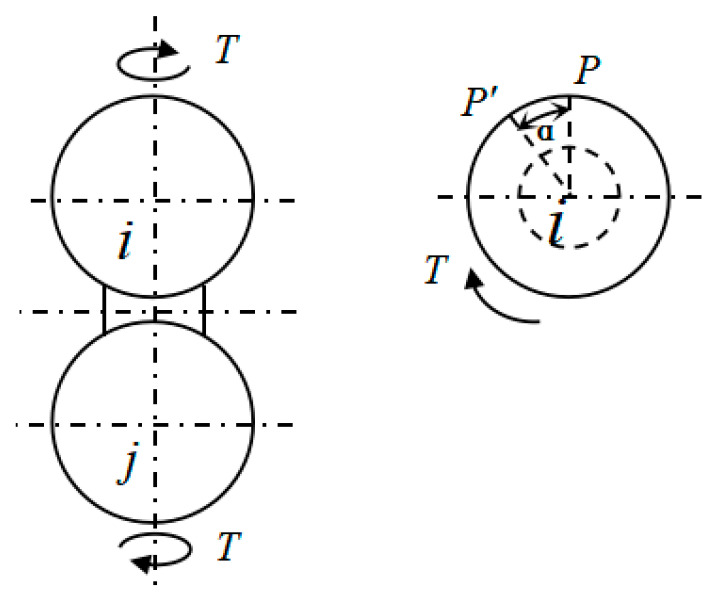
The torsion diagram of a CSE.

**Figure 4 materials-14-03875-f004:**
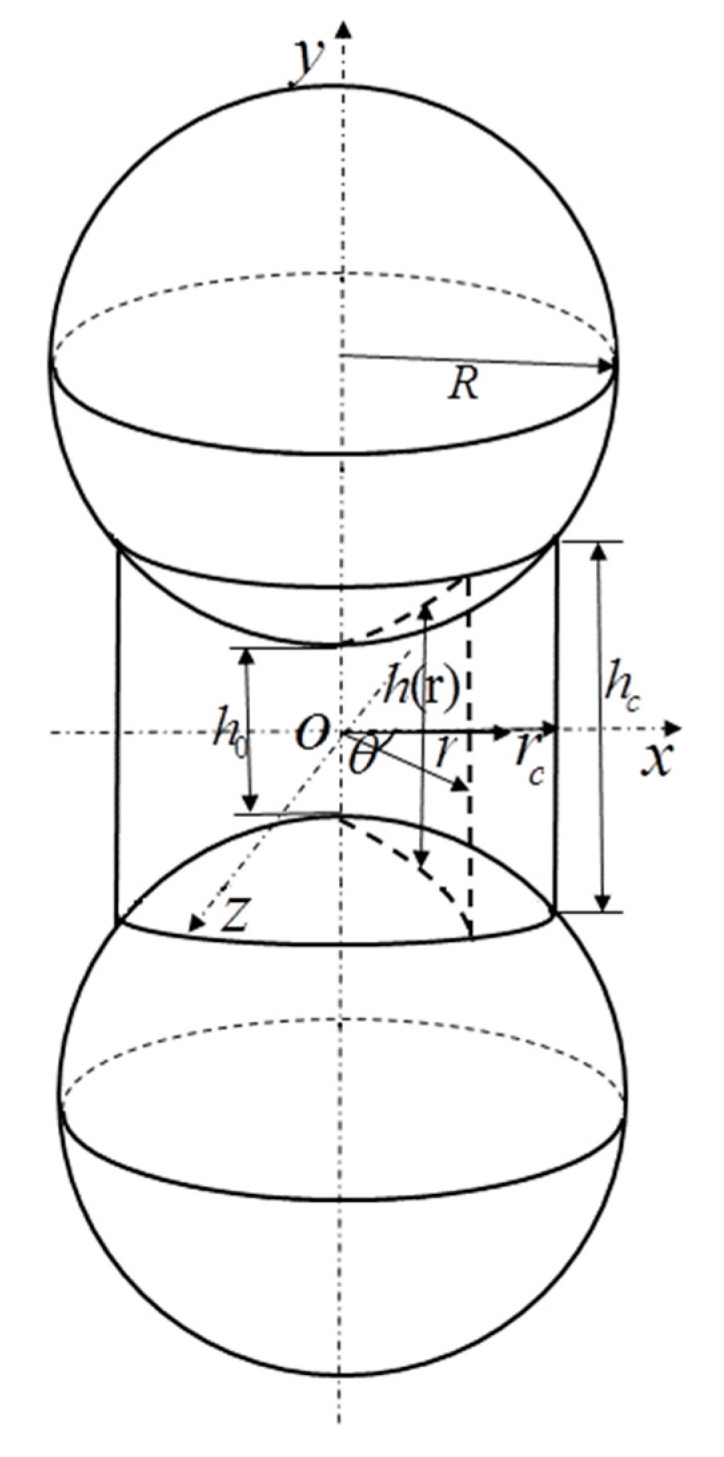
Single CSE coordinate system [31].

**Figure 5 materials-14-03875-f005:**
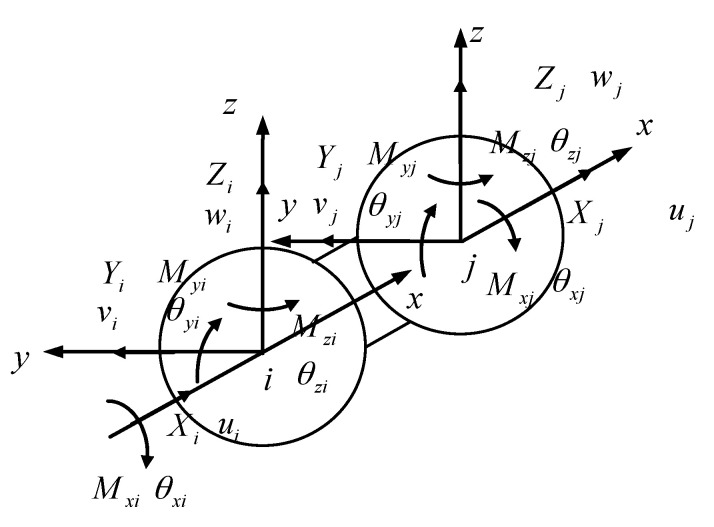
Nodal displacement and nodal force of CSE.

**Figure 6 materials-14-03875-f006:**
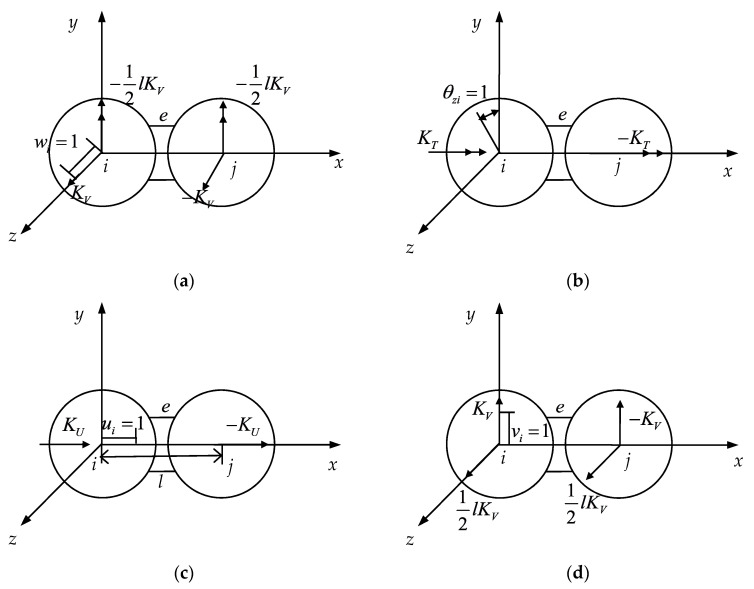
Unit node force caused by unit node displacement.

**Figure 7 materials-14-03875-f007:**
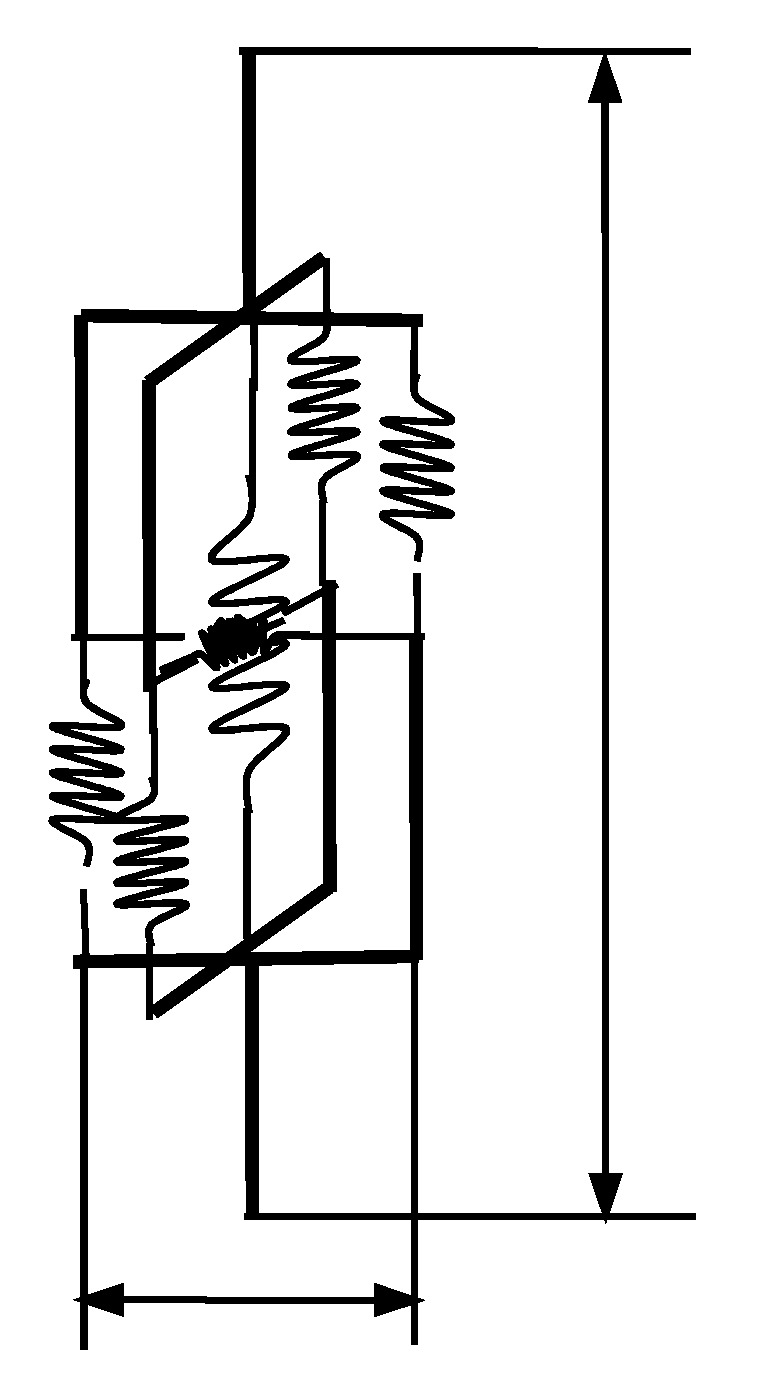
Equivalent rigid beam-spring model.

**Figure 8 materials-14-03875-f008:**
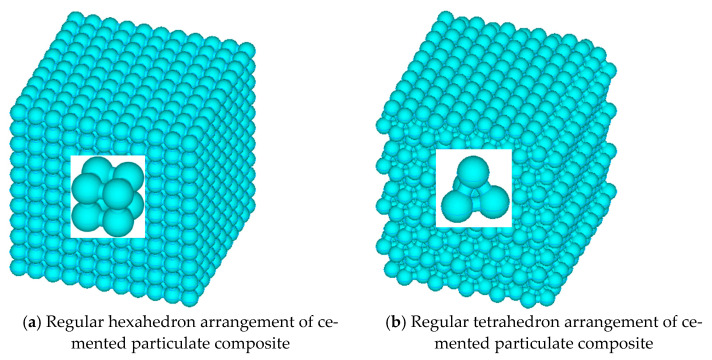
Schematic diagrams of two different arrangement forms of cemented particulate composite.

**Figure 9 materials-14-03875-f009:**
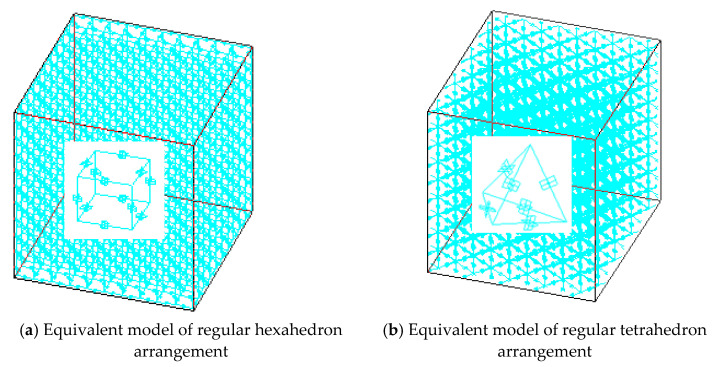
Space equivalent 3D-RBSN model.

**Figure 10 materials-14-03875-f010:**
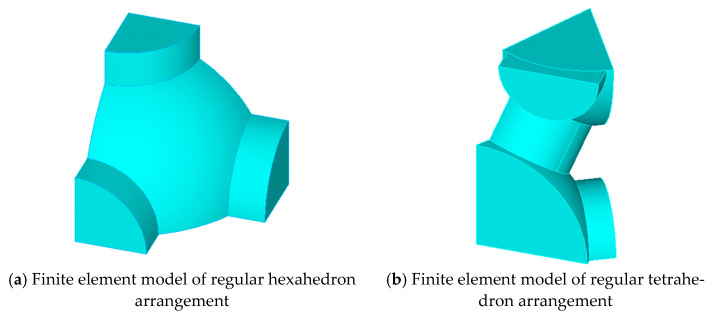
Finite element model of RVE.

**Figure 11 materials-14-03875-f011:**
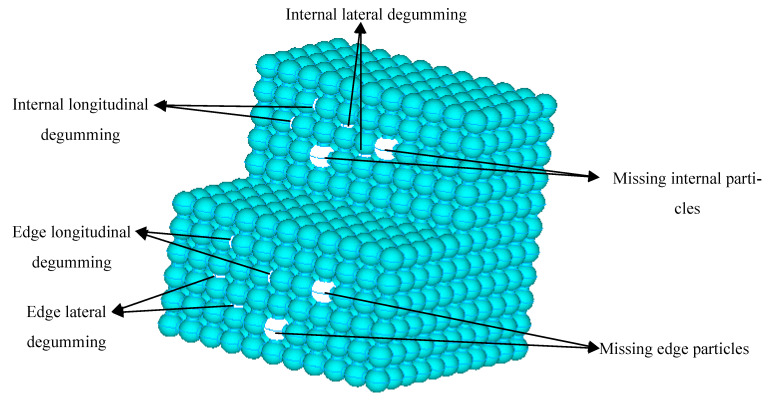
Schematic diagram of material with defective cells.

**Figure 12 materials-14-03875-f012:**
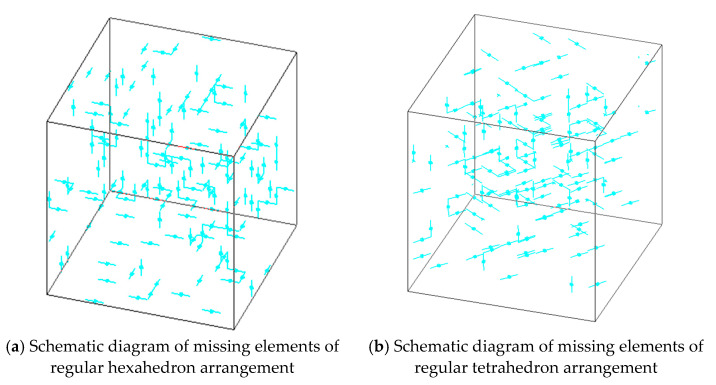
Schematic diagram of missing elements for 3D-RBSN model.

**Figure 13 materials-14-03875-f013:**
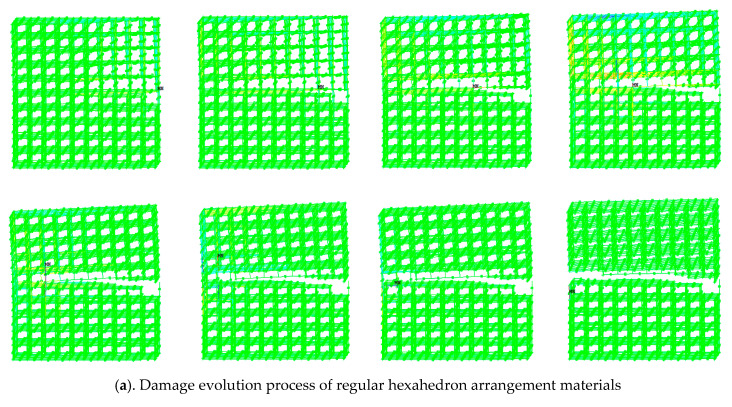
Damage evolution process of materials of two different arrangement forms of cemented particulate composite.

**Table 1 materials-14-03875-t001:** Elastic properties of materials calculated.

Arrangement	E (GPa)	u
Tetrahedron arrangement (CSE)	5.11	0.25
Hexahedron arrangement (CSE)	4.81	0.27
Tetrahedron arrangement (RVE)	5.05	0.26
Hexahedron arrangement (RVE)	4.75	0.28

## Data Availability

Data is contained within the article.

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
