# Peer review of "Mechanical Properties of Cemented Particulate Composite: A 3D Micromechanical Model"

_materials, 2021, doi:10.3390/ma14143875_

Round 1

Reviewer 1 Report

This study investigates the development of a micromechanical model with a three-dimensional rigid beam-spring network. The calculations of the stiffness matrix for the proposed model was obtained and applied to the analysis of composite materials containing glass beads and epoxy resin. The model could effectively predict the properties of the composite material. The damage process of the cemented particulate composite was also analysed.

Please consider the below in improving the quality of the paper.

  • Figure 3: is not clear. Also, the rotational angle must be shown in the figure.
  • Figure 4: it is the same figure mentioned in reference [23]. So, citation must be clear. Also, the symbol A is not shown in in the figure. In addition, other symbols need also to be clarified such as “Alpha” and “Phi”.
  • How did the author obtain the formulas in equation 1 and equation 20?
  • Section 3.1: the shear moduli of resin and glass microspheres are not listed.
  • Page 10: The author needs to show clearly how the model accurately simulates the mechanical properties compared to the existing model in literature (reference [23])
  • Page11: The author mentioned that the mechanical properties with particles arranged in tetrahedron were better than those arranged in hexahedron. Can the author explain how they are better?
  • The author needs to further explain the process of crack initiation in the finite element modelling. Did the author use the same adopted criteria for both arrangements?
  • Editorial and grammatical issues such as:
    • The abbreviation CSE must be defined as “combinational sphere element” where it is firstly mentioned.
    • Page 3: a reference must be provided for the existing calculation method?
    • Section numbering must be corrected
    • Section 2.3 : “ D micro…….”? And, section 2.3.3
    • All figures must be cross-referenced in text, for example Figure 1b, Figure 2b and a-f in Figure 6.
    • The phrase “previous paper” was mentioned few times within the text. It is better to cite it.
    • Equation 17: the symbols must be consistent with the symbols in text.
    • Equation 18: the parameters A and B must be defined.

Reviewer 2 Report

The title of this study is: Mechanical properties of cemented particulate composite:A 3D micromechanical model. In this research work, the Authors based on the theory of CSE, the torsional stiffness of the structure was calculated and the stiffness matrix of 3D CSE was given.

I commented on the manuscript and the comments are presented below:

The Highlights are well.

Part 1: Introduction.

The Introduction to the study is clearly. The purpose of the research was clearly stated.

The Highlights are well.

The structure of the manuscript should include sections such as 2. Materials and Methods and 3. Results and Discussion or separately Result, Discussion. Chapters should be properly numbered. The chapter Materials and methods should contain the necessary information that will enable the repetition and verification of tests. The Results and Discussion section should contain the results of the experiments and their discussion with the results of other researchers, which was not done. In the Discussion chapter, there is no comparison and confrontation with the research of other authors in this area.

Part: 4 Conclusion

The Conclusions chapter contains information obtained after conducting experiments but there are no comparison and confrontation with the research of other authors in this area.

Part: 5 References.

The literature used is appropriate but should be supplementing about the items from the last years of publication about similar problem which should also be used in the discussion of the obtained results. 

Reviewer 3 Report

I recommend accept

Author Response

Thank you very much for your comments on our manuscript.

Reviewer 4 Report

The paper is well structured and clearly written.

The highlights are concise, and they convey the core findings of the article.

The introduction is poor in content and should be extended with a more extensive literature review.

Although the English language is not bad (a revision would be advisable), the manuscript is difficult to follow.

From these reasons, I think that this manuscript, after a minor revision, could be accepted to be published by the Materials journal.

Reviewer 5 Report

Attached!

Round 2

Reviewer 2 Report

The authors referred to the comments from the previous review for the manuscript titled: Mechanical properties of cemented particulate composite:A 3D micromechanical model. I accept explanations. The authors supplemented the discussion with a literature data strengthens the message and importance of information in the manuscript.